# Towards End-to-End Explainable Facial Action Unit Recognition via Vision-Language Joint Learning

## ABSTRACT

Facial action units (AUs), as defined in the Facial Action Coding System (FACS), have received significant research interest owing to their diverse range of applications in facial state analysis. Current mainstream FAU recognition models have a notable limitation, *i.e.*, focusing only on the accuracy of AU recognition and overlooking explanations of corresponding AU states. In this paper, we propose an end-to-end **V**ision-**L**anguage joint learning network for explainable FAU recognition (termed *VL-FAU*), which aims to reinforce AU representation capability and language interpretability through the integration of joint multimodal tasks. Specifically, VL-FAU brings together language models to generate fine-grained local muscle descriptions and distinguishable global face description when optimising FAU recognition. Through this, the global facial representation and its local AU representations will achieve higher distinguishability among different AUs and different subjects. In addition, multi-level AU representation learning is utilised to improve AU individual attention-aware representation capabilities based on multi-scale combined facial stem feature. Extensive experiments on DISFA and BP4D AU datasets show that the proposed approach achieves superior performance over the state-of-the-art methods on most of the metrics. In addition, compared with mainstream FAU recognition methods, VL-FAU can provide local- and global-level interpretability language descriptions with the AUs' predictions.

## CCS CONCEPTS

• **Computing methodologies** → Image representations; **Biometrics**.

## KEYWORDS

Facial action unit recognition, Explainable facial AU recognition, Vision-language joint learning

## 1 INTRODUCTION

Recognized as a fundamental research challenge by the Facial Action Coding System (FACS) [5], facial action unit (FAU) recognition holds significance for analyzing facial states, including facial expression analysis [3], diagnosing mental health issues [32], detecting deception [6], and more. As a result, this area of study has gained increasing interest in recent years.

Early works [20, 42] design hand-crafted features based on the inherent characteristics of facial muscle movements to determine FAU

**Unpublished working draft. Not for distribution.**

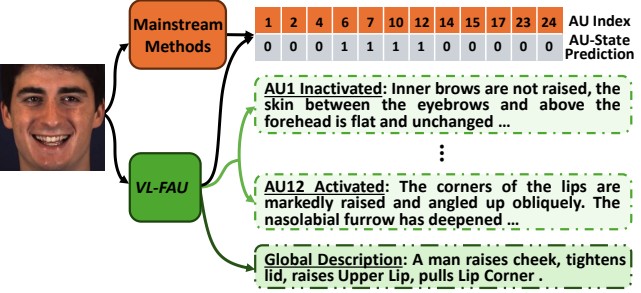

**Figure 1: Comparative analysis of FAU recognition paradigms is shown between conventional methods and our *VL-FAU* model. While the mainstream methods provide direct predictions of AU activation states (orange stream), the VL-FAU model not only offers activation predictions but also provides detailed local and global descriptions of the corresponding AUs in natural language.**

states. However, hand-crafted shallow features prove inadequate to address the major challenges of FAU recognition, i.e. identifying subtle facial changes induced by different AUs, as well as variations stemming from individual physiology. Instead, deep learning based studies [22, 26, 28, 35] have recently been explored to improve the AU representations as a multi-label classification problem, where the AU individual characteristics are adequately learned to boost the FAU recognition. Therefore, improving the individual AU representation becomes the key to boosting the FAU recognition task. The main challenge is how to maintain discriminative inter-AU features while ensuring rich semantic representation capabilities within AUs.

To enhance the individual representation of each AU, mainstream studies [8, 16, 26, 35–39] adopt a multi-branch AU recognition framework with independent classifiers to decouple different AUs. For instance, JAA-Net [34] proposes a face alignment network that predefines local muscle regions for various AUs using detected landmarks. It then refines all AU region representations through multiple independent convolutional networks dedicated to different AU classifications. The predetermined AU local regions and their feature representations establish explicit and distinct branches for conveying information essential for recognizing different AU states. However, they overlook the pertinent information originating from undefined facial areas. Therefore, ARR [37] is proposed to directly leverage a shared global face feature to explore the local AU representations by subsequent independent convolution blocks in different AU branches, covering the whole face information. However, recent studies [7, 16, 23] suggest that the above methods fail to capture intrinsic relationships between independent AU branches, which are actually undeniable [5]. To this end, some works [2, 7, 8, 16, 23] propose to build relationships between predefined AU branches by prior relationship statistics in datasets or implicit relationship reasoning in the modelling networks, such as Long Short-Term Memory network (LSTM) [9], graph convolutional network (GCN), *etc*.

Despite such progress, two prevalent defects remain unexplored: (i) improving the representation capability of individual AUs through inter-AU relational connections may compromise the distinctiveness of features among AUs, and (ii) directly obtaining classification results lacks an explainable basis for judgments. The main reason for the former is that inter-AU information based relationship reasoning networks, such as GCN [16, 38] or GNN [7] *etc.*, are prone to the over-smoothing problem [33] among the different AU nodes, especially on small-scale face datasets. This makes differentiating the AU states difficult. The latter issue is caused by the current mainstream paradigm as shown in Figure 1, which only focuses on the classification results and ignores the corresponding explanations.

In this work, we bring fresh insights toward explainable facial AU recognition via the integration of language generation supervisors within an end-to-end FAU recognition network. We argue that detailed AU language descriptions, such as "The corners of the lips are markedly raised and angled up obliquely. The nasolabial furrow has deepened ..." for activated AU12 (Lip Corner Puller) in Figure 1, can provide more linguistic semantics and relations to corresponding AU appearance features than mainstream relational reasoning methods [7, 16, 26]. Furthermore, different AUs correspond to different descriptions, keeping them better distinguishable. Different from encoding pre-provided activated AU language descriptions into AU classification in [46], we introduce the language generation model as language semantic supervision for the classification of different AU states. It can provide AU prediction with the explainable language descriptions as Figure 1 while providing sufficient semantic to enrich AU representations and supervising the AU detection pertinently to distinguish AU.

Motivated by the above insights, we propose a novel end-to-end vision-language joint learning model (termed *VL-FAU*) for facial AU detection with explainable language generation. Compared with the mainstream complex methods in Figure 1, our main innovations lie in two aspects: (i) exploring the potential of joint language generation auxiliary training for intra-AU semantic enhancement and inter-AU semantic feature differentiation and (ii) providing interpretability for end-to-end FAU recognition. Specifically, we first introduce a new dual-level AU feature refinement based on multi-scale combined representations from a vision backbone. This way, we provide each AU branch with a unique attention-aware representation ability. Secondly, to improve the semantics and guarantee the distinguishability of AU features as well as their interpretability, we integrate the end-to-end multi-branch framework with the local and global language generations for explicit semantic guidance. On one hand, each local branch of AU recognition is simultaneously supervised by a corresponding local language generator via fine-grained semantic-supervised optimization. Such schema constrains the semantics and relationships of AU features by local language modeling and improves the inter-branch distinguishability for each face image. On the other hand, global facial features within and between subjects are difficult to distinguish because muscle changes are mostly subtle under different states. To this end, we introduce a global language model to generate a description focusing on all activated muscles as global semantic supervision. It provides better distinction between different whole-face representations within and between subjects via multiple facial state foci as shown in Figure 1.

Finally, vision AU recognition and language description generation are jointly optimized.

The contributions of this work are as follows:

- We propose a novel end-to-end vision-language joint learning scheme for explainable FAU recognition (VL-FAU), which leverages auxiliary training of language generators to improve discrimination and explanation for FAU recognition.
- We design a new dual-level AU representation learning method based on the multi-scaled facial representation for AU branches, which provides stronger attention-aware AU representation ability;
- We design a novel joint supervision method with local and global language generations for FAU recognition. In such schema, the local language generation provides explicit semantic supervision of each independent AU branch, thereby improving inter-AU discriminability and intra-AU detailed semantics, while the global language model maintains the global distinguishability of different facial states within and between subjects.
- We extend FAU datasets with new local and global language descriptions for different facial muscle states to facilitate language-interpretable FAU recognition.

We conduct extensive experiments on two widely used benchmarks, *i.e.*, BP4D and DISFA, to evaluate the proposed VL-FAU model. VL-FAU outperforms state-of-the-art approaches in AU recognition and provides detailed language descriptions of the individual AU decision and global face state for explanations.

## 2 RELATED WORK

**Facial AU Recognition.** The facial AU recognition task is popularly treated as a multi-label classification problem, utilizing multi-branch independent AU classifiers to predict activation states. From this perspective, most existing methods can be roughly categorized into two groups: region-localization based AU recognition [1, 7, 8, 34, 35, 49] and global-refinement based AU recognition [12, 16, 25, 38, 39]. The former methods represent precise AU representations by local patch feature learning and maintain certain differences between different AU features. For instance, in some early works [41, 48], there is a need to predefine the patch location first. Recent JAA-Net [34] joined AU recognition and face alignment in a unified framework, employing detected landmarks to locate specific AU regions. However, all the above methods focused only on independent regions, without considering non-defined face areas and AU correlations. The latter global-refinement based methods extract AU branch features by directly refining the global face representation to ensure that useful information is not lost. For example, [12] proposed a transformer-based multi-branch AU recognition network, where each branch leverages a shared global face representation and refines it by independent convolution layers. Similarly, the correlations between AUs are also ignored. Recent works [7, 8, 16, 26, 38] pay attention to capturing the relations among AUs for modeling the inherent muscle linkages, supplying information transfers between AU branches. For instance, [16, 23] utilized multiple GCNs to model AU relations, requiring additional pre-statistical AU correlation references from the training set. However, this inter-branch information

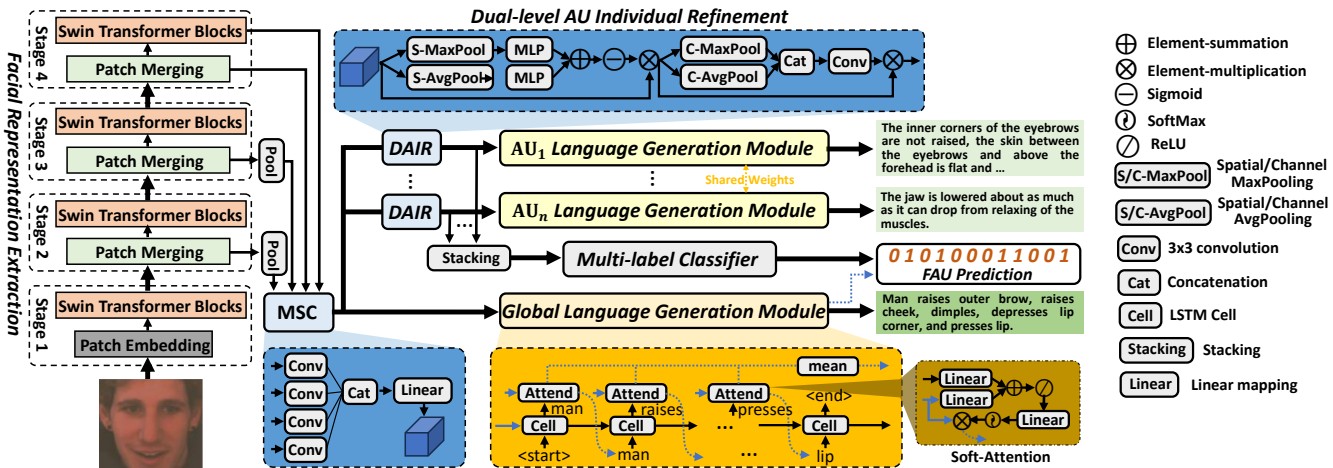

**Figure 2: The overall end-to-end architecture of the proposed VL-FAU for explainable facial AU recognition. Given one face image, the multi-scale combined facial representation is extracted based on a pre-trained Swin-Transformer. VL-FAU is based on the multi-branch network containing multiple independent AU recognition branches as well as a global language generation branch. Each independent AU recognition branch owns a dual-level AU individual refinement module (DAIR) for individual AU attention-aware mining and a local AU language generation module for explicit semantic auxiliary supervision to improve the inter-AU distinguishability. A global language generation based on the multi-scale facial representation is leveraged to preserve shared stem feature diversity via multiple facial state foci. Finally, the multi-branch AU refined representations are stacked for multi-label classification with local and global language auxiliary supervisions (best viewed in color).**

interaction somewhat breaks the original categorical independence between branches, which in turn weakens the discriminability of AU individuals. Moreover, all the above methods are flawed regarding explainable FAU recognition.

**Multi-task Joint Learning.** Recently, multi-task joint learning [8, 11, 31, 50] has been an emerging topic in deep learning research, which enables the same target to efficiently obtain multiple representation capabilities in a unified model. For example, [35] joined face alignment into a facial AU recognition framework to assist in locating the muscles corresponding to AUs. [31] combined four face tasks into a unified framework and assigned an attention module for each face analysis subnet. However, these methods only focus on computer vision tasks and ignore the modality complementary advantages of multiple modality-task joint learning, *i.e.* interpretability complementarity and semantic complementarity. For example, [13] proposed a joint multi-modal aspect-sentiment analysis with auxiliary cross-modal relation detection, which can provide image sentiment predictions with explicit language explanations. However, most visual-language joint learning models [11, 13, 50] rely on the joint input of both modalities during the inference process.

In this paper, we argue that explicit language can facilitate the modeling of inter-AU relevance and diversity. Each AU description can include muscle details and relationship descriptions identified by AU, which not only replace the visual relational reasoning among AUs, but also ensure the independence of AUs to improve distinguishability. The most relevant research to ours is SEV-Net [46], which introduced AU language descriptions as a prior embedding into AU representations. However, significant drawbacks are that SEV-Net relied on the human pre-given language descriptions of

activated AUs for inference and followed the mainstream AU recognition paradigm in Figure 1 for only AU prediction. In contrast to [46], our VL-FAU introduces language generation auxiliary models for local AU prediction and global face refinement. It focuses on end-to-end language-explainable facial-image AU recognition without any pre-given reference for inference.

## 3 APPROACH

As shown in Fig. 2, the proposed approach –VL-FAU– consists of two key components, *i.e.*, multi-level AU representation learning, and local and global auxiliary language generation. The former contains dual-level AU individual refinement for multi-branch FAU recognition based on multi-scale feature combinations from Swin-Transformer. The latter consists of local AU language generation, which facilitates semantic supervision of each AU and global facial language generation for whole-face semantic supervision. Both of these can help AU representations become more robust and distinguishable, thus improving the performance. Thus, we create an end-to-end framework with a multi-label classifier for explainable FAU recognition, supervised with local and global AU language generation auxiliaries.

### 3.1 Multi-level AU Representation Learning

*3.1.1 Global Facial Representation Extraction.* Given a face image $I$, we adapt the widely-used Swin-Transformer as the stem network [24] to extract the global feature $V$ by combining the multi-scale representations from different stages. Multi-scale feature learning and combination [16, 29–31] is a popular image representation approach to leverage different levels of semantics from the backbone, where the low-level features from shallow blocks contain more

texture information and the deeper blocks contain high-level semantics. In the work, we follow this strategy (*multi-scale combination – MSC*) to represent the global face image, which contains 4 independent 3×3 convolutions to learn and reshape the representations $(V_1^s, V_2^s, V_3^s, V_4^s)$ from different stem stages. After that, four-level feature maps are combined as the global facial representation $V$ by a learnable Linear layer. The above process can be formulated as:

$$V = W^m([\text{Conv}(V_1^s) : \text{Conv}(V_2^s) : \text{Conv}(V_3^s) : \text{Conv}(V_4^s)]), \quad (1)$$

where $[:,:]$ means the concatenation operation, $W^m \in \mathbb{R}^{D \times d}$ is mapping parameter and Conv means 3×3 convolution. For simplicity, we will not repeat the detailed structure of the stem Swin-Transformer network [24] here.

### 3.1.2 Dual-level AU Individual Refinement.
Multi-branch network is a good way to learn the rich and fine-grained individual facial AU representation for the final classification. In this paper, we propose a new multi-branch dual-level AU individual refinement (termed *DAIR*) for the final attention-aware AU representations. As shown at the top of Figure 2, each DAIR in each branch contains two levels of attention learning, i.e. channel-level and spatial-level, which employs two pooling strategies [21, 40]. While existing research from both perspectives is extensive [7, 25, 31, 44], this study represents the first adaptation within independent FAU branches for enhancing the independent fine-grained attention-aware AU representations rather than coarse-grained global representation. Specifically, following [44], we extract the max-pooled and average-pooled channel-level vectors ($F_{max}^c$ and $F_{avg}^c$) along the spatial axis for channel-wise attention mining. After that, two shared learnable multi-layer perceptrons (MLP) are used to obtain the mapping of channel-wise vectors and then a sigmoid function is applied to get the $i$-th individual AU-channel attention. Similarly, we later extract the max-pooled and average-pooled spatial-level vectors ($F_{max}^s$ and $F_{avg}^s$) along the channel axis for spatial-wise attention mining. We utilise a 3×3 convolution to generate a spatial attention map, which focuses on highly responsive muscle areas associated with the current AU. Finally, the $i$-th individual AU attention-aware representation $\hat{V}_i$ can be obtained as below:

$$\bar{V}_i = \sigma(\text{MLP}(F_{max}^c) + \text{MLP}(F_{avg}^c))V, \quad (2)$$

$$\hat{V}_i = \text{Conv}([F_{max}^s : F_{avg}^s])\bar{V}_i, \quad (3)$$

where $\sigma$ is the sigmoid function.

## 3.2 Auxiliary Supervision with Local and Global Language Generation

In addition to improving AU individual representations in multiple AU branches, our main innovation is to provide a new approach and inspiration for explainable FAU recognition, *i.e.*, joint auxiliary language generation for FAU recognition, rather than a new complex language model. Language generation provides explicit semantic supervision while giving linguistic interpretability of the corresponding identified AUs, rather than just simply recognising the AUs. It contains two aspects: (i) global face language generation for explicit semantic auxiliary supervision of the whole face image representation, and (ii) local AU language generation for individual AU semantic auxiliary supervision.

### 3.2.1 Global Language Generation .
Different from the mainstream facial AU recognition [16, 34], we introduce a new global language auxiliary model to generate the language description for activated AUs of the whole face. It brings benefits for subsequent multi-branch FAU recognition, i.e., it maintains the accuracy and variability of intra- and inter-subject stem features by explicitly focusing on multiple different AU muscles. Thus, it somewhat overcomes the data imbalance, *i.e.*, inactive AUs are far more numerous than activated AUs in benchmarks.

Specifically, as shown in Figure 2, we treat the multi-scale stem-feature $V$ as the encoded image feature and input it into an attention-aware language decoder similar to image captioning [45], which contains a soft-attention module to mine the attention-aware visual representation based on the past generated word $s_{0:i-1}^g$ for new word $s_i^g$. For example, when generating current word $s_i^g$, we first calculate the soft-attention $\alpha_i$ between the image feature $V$ and the last hidden state $h_{i-1}$ of generated word $s_{i-1}^g$ by linear-based mapping operations with SoftMax function [15]. And then the $i$-th attention-aware visual feature $\alpha_i V$ is combined with the last hidden state $h_{i-1}$, which is fed into the $i$-th LSTM cell [10] with the previous cell state $c_{i-1}$ to generate the new hidden state $h_i$ and cell state $c_i$ of new word. The above process can be formulated as:

$$\alpha_i = \text{SoftMax}(W^a(\text{ReLU}(W^v V + W^h h_{i-1}))), \quad (4)$$

$$(h_i, c_i) = \text{Cell}([\alpha_i V : h_{i-1}], c_{i-1}), \quad (5)$$

where $W^v \in \mathbb{R}^{d \times d}$, $W^h \in \mathbb{R}^{h \times d}$, and $W^a \in \mathbb{R}^{1 \times d}$ are the parameters of mapping function. During the training process, we use a shared learnable parameter $W^s \in \mathbb{R}^{h \times voc}$ to obtain vocabulary-length predicted vector and obtain the max-score index as the predicted word, as follows:

$$s_i^g = \arg\max(W^s h_i), \quad (6)$$

During inference process, the beam search [14] can be used to obtain the most optimal global description $S^g = [s_1^g, ..., s_T^g]$.

### 3.2.2 Local Language Generation.
We introduce an individual local language generation model for each AU branch as an auxiliary semantic supervision, which can generate the corresponding fine-grained language description for each AU determination. Compared with the global facial description, the local AU language description contains more facial muscle details described for corresponding AUs. Besides, different AUs have different local language descriptions, which are more fine-grained and diverse. The main motivation of the joint language model in each AU branch is not only to improve the distinguishability between AUs using fine-grained semantic auxiliary supervision, but also to provide specific language interpretation for each AU prediction.

In particular, different from global language generation, each local language model utilizes the proposed DAIR to further refine the encoded face feature $V$, obtaining the distinguishable attention-aware AU representation $\hat{V}_i$ for the subsequent language generation. The decoder architecture of each local language generation model is the same as the global language model. To save space, we use $\phi_i$ to represent the $i$-th local language model for the $i$-th AU branch and omit the model details. Finally, we can obtain the $i$-th local AU

**Table 1: Comparisons of AU recognition for 8 AUs on DISFA in terms of F1-frame score (in %).**

| Method | AU Index | | | | | | | | Avg. |
|---|---|---|---|---|---|---|---|---|---|
| | 1 | 2 | 4 | 6 | 9 | 12 | 25 | 26 | |
| JAA-Net$_{(ECCV2019)}$ [34] | 43.7 | 46.2 | 56.0 | 41.4 | 44.7 | 69.6 | 88.3 | 58.4 | 56.0 |
| UGN-B$_{(AAAI2021)}$ [38] | 43.3 | 48.1 | 63.4 | 49.5 | 48.2 | 72.9 | 90.8 | 59.0 | 60.0 |
| HMP-PS$_{(CVPR2021)}$ [39] | 21.8 | 48.5 | 53.6 | 56.0 | $\underline{58.7}$ | 57.4 | 55.9 | 56.9 | 61.0 |
| FAU-Trans$_{(CVPR2021)}$ [12] | 46.1 | 48.6 | 72.8 | **56.7** | 50.0 | 72.1 | 90.8 | 55.4 | 61.5 |
| ME-GraphAU$_{(IJCAI2021)}$ [26] | 54.6 | 47.1 | $\underline{72.9}$ | 54.0 | 55.7 | $\underline{76.7}$ | 91.1 | 53.0 | 63.1 |
| KDSRL$_{(CVPR2022)}$ [1] | 60.4 | 59.2 | 67.5 | 52.7 | 51.5 | 76.1 | 91.3 | 57.7 | $\underline{64.5}$ |
| KS$_{(ICCV2023)}$ [18] | 53.8 | **59.9** | 69.2 | $\underline{54.2}$ | 50.8 | 75.8 | $\underline{92.2}$ | 46.8 | 62.8 |
| AAR$_{(TIP2023)}$ [37] | **62.4** | 53.6 | 71.5 | 39.0 | 48.8 | 76.1 | 91.3 | **70.6** | 64.2 |
| SMA-ViT$_{(TAC2023)}$ [19] | 51.2 | 49.3 | 64.7 | 48.3 | 50.6 | **87.6** | 85.1 | 61.2 | 62.2 |
| **VL-FAU(ours)** | $\underline{60.9}$ | $\underline{56.4}$ | **74.0** | 46.3 | **60.8** | 72.4 | **94.3** | $\underline{66.5}$ | **66.5** |
| SEV-Net$_{(CVPR2021)}$ [46] | 55.3 | 53.1 | 61.5 | **53.6** | 38.2 | 71.6 | 95.7 | 41.5 | 58.8 |
| **VL-FAU(ours)** | **60.9** | **56.4** | **74.0** | 46.3 | **60.8** | **72.4** | 94.3 | **66.5** | **66.5** |

description $S_i^l = [s_{1;i}^l, ..., s_{T;i}^l]$ with length $T$, as follows:

$$S_i^l = \oint_i (DAIR_i(V)),\tag{7}$$

Note that, the local language models among the multiple AU branches are shared to maintain efficiency.

## 3.3 Vision-Language Joint Learning

VL-FAU joints facial AU recognition (vision) and description generation (language) into an end-to-end multi-branch network. Among them, FAU recognition is predominant due to the explicit AU annotations, while the language models are used for auxiliary semantic supervision and to improve feature diversity and distinguishability of visual stem and sub-branches. For facial AU recognition, one fully connected layer with SoftMax function is used as a multi-label binary classifier to classify the AU activation state, which adopts a weighted multi-label cross-entropy loss function as follows,

$$\mathcal{L}_{Fau} = -\frac{1}{N}\sum_{i=1}^{N}\gamma_i[y_i\log(p(y_i)) + (1 - y_i)\log(1 - p(y_i))],\tag{8}$$

$$\gamma_i = \frac{1/\epsilon_i}{\sum_{i=1}^{N}(1/\epsilon_i)}\tag{9}$$

where $N$ is AU number, $y_i$ and $p(y_i)$ denote the ground-truth and predicted probability for the $i$-th AU occurrence, respectively. $\gamma_i$ is a balancing weight of the $i$-th AU calculated by the $i$-th AU occurrence rate $\epsilon_i$ in the training set.

Local AU language generation auxiliary training provides detailed semantic supervision for AU predictions and is optimised by commonly used negative log-likelihood loss, as follows:

$$\mathcal{L}_{Lgen} = -\frac{1}{N}\sum_{i=1}^{N}\frac{1}{T}\sum_{t=1}^{T}(\log p(s_{t;i}^l|\hat{V}_i, s_{[0:t-1];i}^l))\tag{10}$$

where N is the number of AU branches and T is the length of each description. The $t$-th word $s_{t;i}^l$ of $AU_i$ description is generated based on the previous words $s_{[0:t-1];i}^l$ and the $AU_i$ refined visual feature $\hat{V}_i$.

Moreover, the introduced global language generation is also optimized by a similar objective function in Eq. (10) with the global generation $S^g = [s_1^g, ..., s_T^g]$ as $\mathcal{L}_{Ggen}$:

$$\mathcal{L}_{Ggen} = -\frac{1}{T}\sum_{t=1}^{T}(\log p(s_t^g|V, s_{[0:t-1]}^g))\tag{11}$$

Different from the local language generation for specific AU branches, the global language generation faces the challenge of linguistic semantic diversity due to facial state changes. To this end, we add a global AU classification loss $\mathcal{L}_{Gau}$ together with $\mathcal{L}_{Ggen}$ as a constraint, where AU states are predicted by a shared linear classifier based on the average attention-aware visual feature from language model.

Finally, the joint loss of our VL-FAU model can be optimized by maximizing the following lower bound:

$$\mathcal{L} = \mathcal{L}_{Fau} + \mathcal{L}_{Lgen} + \mathcal{L}_{Ggen} + \mathcal{L}_{Gau}.\tag{12}$$

## 4 EXPERIMENTS

### 4.1 Dataset and Implementation Details

**Dataset.** We provide evaluations on the widely used datasets, *i.e.* BP4D [47] and DISFA [27]. **BP4D** is a spontaneous FAU database containing 328 facial videos from 41 subjects (23 females and 18 males) who were involved in 8 sessions. Similar to [35, 36], the most expressive frames are manually labeled with AU occurrence from each session (around 400-500 frames). Overall, BP4D contains 12 AUs and 140K valid frames with labels. **DISFA** consists of 27 participants (12 females and 15 males). Each participant has a video of 4, 845 frames. 8 AU annotations are selected following the popular studies [17, 35]. Compared to BP4D, the experimental protocol and lighting conditions deliver DISFA to be a more challenging dataset. **Data Processing**[1]. During training, language descriptions of different AU states are hand-crafted, containing descriptions of both activated and inactivated AU states. Each local AU description contains multiple muscle details with potential associations according to FACS [5]. The global face description is generated from the AU annotations and FACS [5], which only focus on the activated AU muscles. The examples are shown in Figure 5. We evaluated the model using the common 3-fold subject-exclusive cross-validation protocol [8, 19, 34, 35].

**Training strategy.** The whole end-to-end network is implemented with PyTorch on a single NVIDIA RTX 3090Ti GPU using AdamW solver with $\beta_1 = 0.9$, $\beta_2 = 0.999$, and a weight decay of 0.0005. Maximum epochs are set to 15 with a batch size of 64. During training process, aligned faces are randomly cropped into $224 \times 224$ and

---

[1]The details of language descriptions will be released with paper acceptance.

**Table 2: Comparisons with state-of-the-art methods for 12 AUs on BP4D in terms of F1-frame(in %).**

| Method | 12 AUs | | | | | | | | | | | | Avg. |
|---|---|---|---|---|---|---|---|---|---|---|---|---|---|
| | 1 | 2 | 4 | 6 | 7 | 10 | 12 | 14 | 15 | 17 | 23 | 24 | |
| JAA-Net(ECCV2019) [34] | 47.2 | 44.0 | 54.9 | 77.5 | 74.6 | 84.0 | 86.9 | 61.9 | 43.6 | 60.3 | 42.7 | 41.9 | 60.0 |
| LGRNet (FG2021) [8] | 50.8 | 47.1 | 57.8 | 77.6 | 77.4 | 84.9 | 88.2 | 66.4 | 49.8 | 61.5 | 46.8 | 52.3 | 63.4 |
| UGN-B (AAAI2021) [38] | 54.2 | 46.4 | 56.8 | 76.2 | 76.7 | 82.4 | 86.1 | 64.7 | 51.2 | 63.1 | 48.5 | 53.6 | 63.3 |
| HMP-PS(CVPR2021) [39] | 53.1 | 46.1 | 56.0 | 76.5 | 76.9 | 82.1 | 86.4 | 64.8 | 51.5 | 63.0 | 49.9 | 54.5 | 63.4 |
| FAU-Trans(CVPR2021) [12] | 51.7 | 49.3 | 61.0 | 77.8 | 79.5 | 82.9 | 86.3 | 67.6 | 51.9 | 63.0 | 43.7 | 56.3 | 64.2 |
| ME-GraphAU(IJCAI2022) [26] | 53.7 | 46.9 | 59.0 | 78.5 | 80.0 | 84.4 | 87.8 | 67.3 | 52.5 | 63.2 | 50.6 | 52.4 | 64.7 |
| KDSRL(CVPR2022) [1] | 53.3 | 47.4 | 56.2 | 79.4 | 80.7 | **85.1** | **89.0** | 67.4 | **55.9** | 61.9 | 48.5 | 49.0 | 64.5 |
| KS(ICCV2023) [18] | 55.3 | 48.6 | 57.1 | 77.5 | **81.8** | 83.3 | 86.4 | 62.8 | 52.3 | 61.3 | 51.6 | 58.3 | 64.7 |
| AAR(TIP2023) [37] | 53.2 | 47.7 | 56.7 | 75.9 | 79.1 | 82.9 | 88.6 | 60.5 | 51.5 | 61.9 | 51.0 | 56.8 | 63.8 |
| SMA-ViT(TAC2023) [19] | 52.7 | 45.6 | 59.8 | **83.8** | 79.2 | 83.5 | 87.2 | 64.0 | 54.1 | 61.2 | **52.6** | 58.3 | 65.2 |
| **VL-FAU(ours)** | **56.3** | **49.9** | **62.6** | 79.5 | 80.1 | 82.6 | **88.6** | 66.8 | 51.3 | **63.5** | 51.3 | 57.1 | **65.8** |
| SEV-Net(CVPR2021) [46] | 58.2 | 50.4 | 58.3 | **81.9** | 73.9 | **87.8** | 87.5 | 61.6 | **52.6** | 62.2 | 44.6 | 47.6 | 63.9 |
| **VL-FAU(ours)** | **56.3** | **49.9** | **62.6** | 79.5 | **80.1** | 82.6 | **88.6** | **66.8** | 51.3 | **63.5** | **51.3** | **57.1** | **65.8** |

**Table 3: Comparisons with state-of-the-art methods on DISFA and BP4D in terms of Accuracy(in %).**

| Method | 8 AUs (DISFA) | | | | | | | | Avg. | 12 AUs (BP4D) | | | | | | | | | | | | Avg. |
|---|---|---|---|---|---|---|---|---|---|---|---|---|---|---|---|---|---|---|---|---|---|---|
| | 1 | 2 | 4 | 6 | 9 | 12 | 25 | 26 | | 1 | 2 | 4 | 6 | 7 | 10 | 12 | 14 | 15 | 17 | 23 | 24 | |
| JAA-Net(ECCV2019)[34] | 93.4 | 96.1 | 86.9 | 91.4 | 95.8 | 91.2 | 93.4 | 93.2 | 92.7 | 74.7 | 80.8 | 80.4 | 78.9 | 71.0 | **80.2** | 85.4 | 64.8 | 83.1 | 73.5 | 82.3 | 85.4 | 78.4 |
| JÂANet(IJCV2021)[35] | **97.0** | **97.3** | 88.0 | 92.1 | 95.6 | 92.3 | 94.9 | **94.8** | **94.0** | 75.2 | 80.2 | 82.9 | 79.8 | 72.3 | 78.2 | 86.6 | 65.1 | 81.0 | 72.8 | **82.9** | 86.3 | 78.6 |
| UGN-B(AAAI2021)[38] | 95.1 | 93.2 | 88.5 | **93.2** | **96.8** | **93.4** | 94.8 | 93.8 | 93.4 | 78.6 | 80.2 | 80.0 | 76.6 | 72.3 | 77.8 | 84.2 | 63.8 | **84.0** | 72.8 | 82.8 | **86.4** | 78.2 |
| **VL-FAU(ours)** | 96.5 | 96.9 | **92.0** | 91.0 | 96.3 | 91.8 | **96.7** | 93.0 | **94.3** | **79.1** | **82.3** | **83.0** | **80.5** | **77.4** | 78.7 | **86.8** | 64.9 | 82.9 | 73.4 | 82.6 | 86.3 | **79.8** |

horizontally flipped. We randomly employ the cutout augmentation [4] to overcome overfitting and improve the robustness during training. The stem extractor (Swin-Transformer-base) is pre-trained on ImageNet. The dimensionality of the global feature is mapped from 1024 to 512 (d=512), matching the hidden state dimension (h=512).

## 4.2 State-of-the-art Comparisons

We perform extensive experiments to compare our VL-FAU with mainstream FAU recognition studies and the latest state-of-the-art methods on two widely used FAU benchmarks in Table 1 and Table 2. The best and second-best results are bold and underlined.

**Evaluation metrics.** For all methods, the frame-based F1 score (F1-frame, %) is reported, which is the harmonic mean of the Precision P and Recall R and calculated by F1 = 2PR/(P+R). To conduct a more comprehensive comparison with other methods, we also evaluate the accuracy performance (%). In addition, the average results over all AUs (denoted as **Avg.**) are computed with "%" omitted.

**Quantitative comparison on DISFA:** We compare our proposed VL-FAU with its counterpart in Table 1 and Table 3. Our VL-FAU outperforms mainstream studies with impressive margins. In particular, compared with the state-of-the-art AAR [37], which joint surprised local attention maps to obtain the multi-branch attention-aware AU representation, our VL-FAU increases the average F1-frame by 2.3% and shows clear improvements for most annotated AU categories. Compared with the best model KDSRL [1] on DISFA, the average F1-frame score of our VL-FAU is also improved from 64.5% to 66.5%. Furthermore, we achieve the best performance in terms of average accuracy in Table 3, compared with all methods.

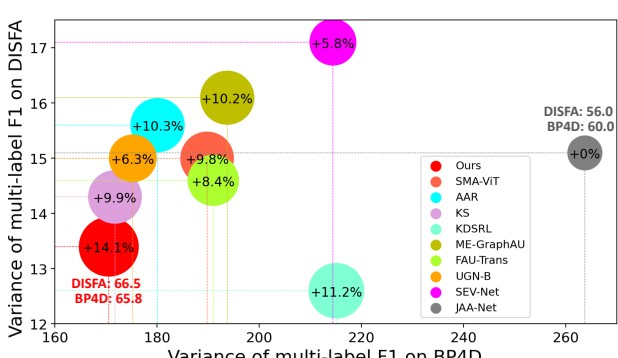

**Figure 3: Multi-Label Performance Balancing Analysis. X-axis and Y-axis denote the variances of multi-label F1 scores on BP4D and DISFA, respectively. Circle size indicates the total relative performance improvement (%) compared with JAA-Net on BP4D and DISFA.**

**Quantitative comparison on BP4D:** FAU recognition results by different methods on BP4D are shown in Table 2 and Table 3, where the proposed VL-FAU model achieves a new state of the art compared with all methods in terms of average F1-frame score. Our VL-FAU outperforms the baseline JAA-Net [34], which integrates AU detection and face alignment, in terms of average F1-frame and accuracy by 5.8% and 1.4%, respectively. This is mainly because JAA-Net focuses on the local AU regions based on the detected landmarks, resulting in poor distinguishability between different AUs, especially for the same individual with subtly different face states, which can be improved by our method. Moreover, VL-FAU achieves the best

**Table 4: Effectiveness of key components of VL-FAU evaluated on BP4D in terms of F1-frame score (in %) of FAU recognition and top-5 accuracy (in %) of local and global language generation models.**

| Model | | Setting | | | AU Index | | | | | | | | | | | | Avg. | LLGA Acc. | GLGA Acc. |
|---|---|---|---|---|---|---|---|---|---|---|---|---|---|---|---|---|---|---|---|
| | | MARL | LLGA | GLGA | 1 | 2 | 4 | 6 | 7 | 10 | 12 | 14 | 15 | 17 | 23 | 24 | | | |
| VL-FAU | ① | - | - | - | 50.0 | 46.3 | 60.3 | 78.0 | 80.0 | 83.8 | 88.6 | 64.1 | 50.2 | 64.0 | 50.1 | 56.5 | 64.3 | - | - |
| | ② | √ | - | - | 50.0 | 45.5 | 60.0 | 79.6 | 80.1 | 83.1 | 88.7 | 66.5 | 51.0 | 63.3 | 53.6 | 55.3 | 64.7 | - | - |
| | ③ | √ | √ | - | 51.4 | 48.2 | 60.2 | 79.1 | 80.8 | 83.4 | 88.6 | 65.0 | 52.4 | 65.6 | 52.1 | 57.0 | 65.3 | 86.3 | - |
| | ④ | √ | - | √ | 54.8 | 47.4 | 61.2 | 79.2 | 79.4 | 84.1 | 88.8 | 63.8 | 52.2 | 65.2 | 50.6 | 55.6 | 65.2 | - | 64.4 |
| | ⑤ | √ | √ | √ | 56.3 | 49.9 | 62.6 | 79.5 | 80.1 | 82.6 | 88.6 | 66.8 | 51.3 | 63.5 | 51.3 | 57.1 | **65.8** | 86.6 | 64.7 |

or second-best F1 and accuracy scores in recognizing most of the 12 AUs in BP4D, outperforming other state-of-the-art methods.

In addition, compared with SEV-Net [46], which prior encodes the pre-provided linguistic descriptions into image features, our VL-FAU achieves 7.7% and 1.9% higher average F1-frame scores on DISFA and BP4D, respectively. Experimental results demonstrate the effectiveness of VL-FAU in improving AU recognition accuracy on DISFA and BP4D by our proposed joint learning with language generation. Besides, as shown in Figure 3, we also provide the multi-label performance balancing analysis on DISFA and BP4D. Although the performance varies greatly between different AUs due to the inherent category imbalance in datasets, our VL-FAU still achieves a better performance-balance than existing methods and maintains the best overall results.

## 4.3 Ablation Studies

We perform extensive ablation studies on BP4D to investigate how each component affects the overall performance of the proposed VL-FAU. Due to space limitations, we do not show the ablation results for DISFA, but it is consistent with BP4D. Table 4 presents component ablation studies focusing on the various modules within VL-FAU, including (1) multi-level AU representation learning (MARL), (2) local language generation auxiliary (LLGA), and (3) global language generation auxiliary (GLGA). In addition, Figure 4 gives a qualitative analysis of local and global language generations.

**(1) Multi-level AU Representation Learning (MARL).** Compared with the baseline model ①, the result has an improvement of average F1-frame from 64.3% to 64.7% when considering the proposed multi-level AU representation learning (indicated variant ② with MARL). It indicates that the dual-level individual AU refinement based on the multi-scale stem feature combination could get richer and more fine-grained AU features and hence improve recognition performance.

**(2) Local Language Generation Auxiliary (LLGA).** In Table 4, we test the contributions of local language generation auxiliary (LLGA) of our VL-FAU model. Compared with variant ② and variant ③, after we provide local language generation auxiliary for each AU branch, the average F1-frame score has been improved from 64.7% to 65.3%. In addition, most of the 12 AUs annotated in BP4D achieve significant improvements. These observations demonstrate that by providing each AU branch with local language generation as an explicit auxiliary semantic supervision, the discriminative ability between AUs becomes stronger due to the gain of language expressiveness rather than single visual appearance features.

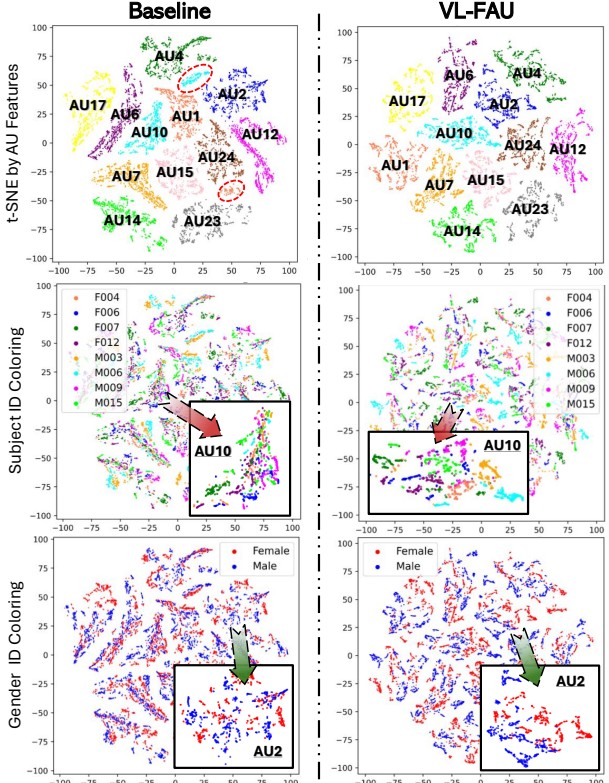

**Figure 4: t-SNE visualization of the baseline model (w/o local and global language generation auxiliary) and full VL-FAU mdoel on BP4D.**

**(3) Global Language Generation Auxiliary (GLGA).** Similarly, we conduct a comparison between variant ② and variant ④ to verify the effectiveness of the proposed global language generation auxiliary (GLGA) for whole-face representation learning. Results in Table 4 show that the proposed VL-FAU facilitates the target FAU recognition task when using GLGA (indicated variant ④). In particular, the averaged F1-frame score increases from 64.7 % to 65.2 % and most AUs achieve significant improvements. These validate the effectiveness of the proposed GLGA which provides better discriminability of global representations by focusing on the language semantics of activated facial AUs, especially for the same subject with subtly different AU states.

Finally, when both local and global language generation auxiliaries (indicated model ⑤) are considered, the proposed VL-FAU achieves the best recognition performance in terms of average F1, significantly better than the single variants (variant ③ and ④) and variant ② in Table 4. In addition, the local and global language generation performances also have certain improvements, achieving 86.6% and 64.7% on top-5 accuracy of word generation in LLGA and GLGA. These quantitative comparisons experimentally demonstrate that exploring explicit semantic-auxiliary supervisions for facial AU recognition is a beneficial way for discriminating different AU states under intra- and inter-subject.

**(4) Qualitative Analysis of Local and Global Generation Auxiliary.** Besides the above quantitative comparisons, we further proved the detailed qualitative analysis of our main innovations – local and global generation auxiliaries for facial AU recognition. As shown in Figure 4, we use t-SNE [43] to visualize the AU features learned by the proposed VL-FAU and the corresponding baseline without local and global generation auxiliaries (baseline ②). Note that, both models are used the same multi-branch networks with MARL. Specifically, we extract the AU features of 8 random gender-balanced subjects for clearer visualization from multiple AU branches before the final classification and then visualize them by t-SNE. We provide different coloring schemes in Figure 4 to analyze the impacts of VL-FAU on different aspects, including AU category, subject ID, and gender. (1) Comparisons of clustered AU features in the first row, our VL-FAU can better divide different AU features into different clusters, while the baseline is not sufficiently discriminative on some AU features (marked with red circles). Besides, we notice that AU features optimized from baseline are mapped in a narrow space compared with our VL-FAU. These observations indicate that by joining language generation auxiliary, our VL-FAU can maintain higher discriminability between multiple AU representations. (2) The second row shows the subject ID coloring results. Our VL-FAU distinguishes different subject AU features more widely within the same AU cluster, indicating that our VL-FAU provides higher discriminability between different subjects for facial AU recognition. (3) Due to the explicit language supervision of gender information in GLGA, VL-FAU can achieve clearer gender colorization results, as shown in the last row of Figure 4.

### 4.4 Visualization of Results

To further understand the quality of the proposed vision-language joint learning for FAU recognition and description generation, we visualize the predicted AU states and their corresponding local and global-level descriptions, as shown in Figure 5. Two positive examples from BP4D contain visualizations of different genders with different AU states. Compared with the mainstream paradigm, our VL-FAU provides explainable FAU recognition with language generations. In detail, local descriptions contain multiple detailed muscle changes with natural connections, improving intra-AU semantics and inter-AU distinguishability. In addition, the global descriptions contain diverse activated AU states with gender information, which can improve the inter-face distinguishability within and between subjects. Besides, we provide a bad case, which makes wrong predictions in two AUs. However, the global description ignores the misrecognition. Although AU6 is incorrectly predicted, the detailed

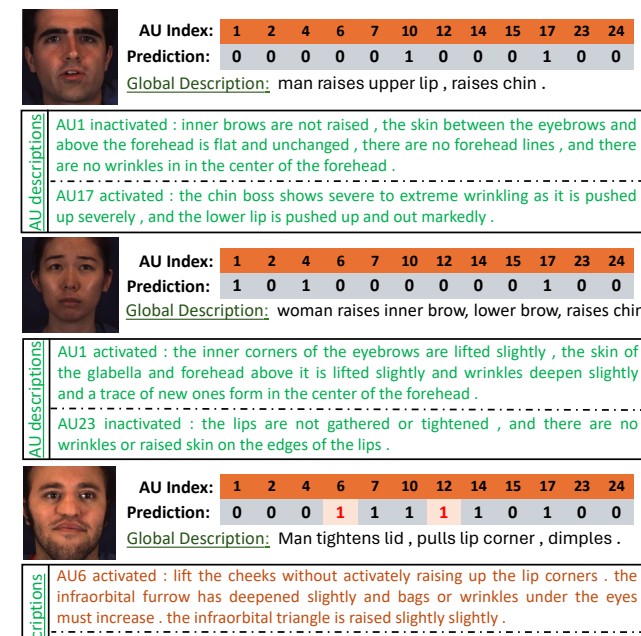

**Figure 5: Visualizations of our proposed explainable facial AU recognition (VL-FAU) with explicit local and global language descriptions on BP4D.**

description matches the facial expression, possibly due to labeling ambiguity. Overall, our VL-FAU can give better explainable facial AU recognition with explicit local and global language descriptions.

## 5 CONCLUSION AND FUTURE WORK

In this paper, we proposed a novel end-to-end vision-language joint learning (VL-FAU) for explainable FAU recognition along with language generations as explanations. As auxiliary supervisions, local and global language generations are joined into a multi-branch AU recognition network with multi-level AU representation learning. Local AU language generation provides explicit fine-grained semantic supervision for each AU classification with detailed language descriptions, improving the discrimination of inter-AU representations. Global language generation, employing multi-scale combined stem features, offers diverse semantic supervision for the whole facial feature to maintain diversity and distinction across intra- and inter-subject representation changes. Our VL-FAU finally provides predictions of AU states as well as interpretable language descriptions for individual AUs and global faces. Extensive experimental evaluations on DISFA and BP4D show that our VL-FAU outperforms state-of-the-art AU recognition methods with impressive margins.

VL-FAU introduces a traditional language model for explainable FAU recognition considering computational power and efficiency limitations. We believe our attempts provide new inspiration for multimodal multi-task joint training for explainable FAU recognition. In the future, we would like to investigate the further combination of FAU recognition with popular LLMs for more diverse and fine-grained explainable generations.

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
