# OpenReview forum: "Towards End-to-End Explainable Facial Action Unit Recognition via Vision-Language Joint Learning"
_acmmm.org/ACMMM/2024/Conference — MM2024 Poster_

### Official Review · Reviewer_RifT · 2024-04-27
**This paper utilizes language description to obtain more detailed AU detection, which is useful. But there are still some questions for speed and experimental designs.**

**Rating:** 5
**Confidence:** 3

**Review:**

Pros:
1. This work achieves explainable AU detection by integrating language into the AU detection task. It changes a binary classification task into a caption generation task and classification task.
2. The performance seems great.

Cons:
1. The choice of the language model. How about using the pre-trained large language model to generate captions and help the classification task?
2. How about training time and inference speed?
3. Why does AU6 perform well in the BP4D dataset but badly in the DISFA dataset?

**Summary:**

This paper utilizes language description to obtain more detailed explainable AU detection, which is useful and novel. However, there are still some questions about inference speed and detailed experimental designs.

**Strengths:**

1. This work achieves explainable AU detection by ingeniously integrating language into the AU detection task. By transforming a traditional binary classification task into a combined caption generation and classification task, it not only identifies AUs but also provides textual descriptions, enhancing the interpretability and transparency of the model's predictions. This novel approach opens avenues for deeper understanding and analysis of facial expressions.
2. The performance metrics indeed indicate remarkable results. The robustness and accuracy demonstrated in the experimental outcomes underscore the effectiveness of the proposed methodology. Additionally, the integration of language not only enhances performance but also enriches the output by providing descriptive captions alongside AU classifications, making the results more informative and actionable.

**Limitations:**

1. The choice of the language model. How about using the pre-trained large language model to generate captions and help the classification task?
2. How about training time and inference speed? Does the caption generation process cost too much?
3. Why does AU6 perform well in the BP4D dataset but badly in the DISFA dataset?

**Suitability:**

3

---

### Official Review · Reviewer_wNWr · 2024-05-19

**Rating:** 4
**Confidence:** 3

**Summary:**

This paper proposes a new end-to-end vision-language joint learning model called VL-FAU for explainable facial action unit (AU) recognition. This paper would fit well under the theme "Multimedia Content Understanding" and "Vision and Language"

**Strengths:**

Integrating auxiliary language generation tasks with the AU recognition network to improve representation capability and provide linguistic interpretability is a unique approach. The use of local language generation modules for individual AU branches and a global language module for the whole face representation is an innovative idea to enhance discriminability and interpretability.

The paper provides a sound theoretical basis for the vision-language joint learning by leveraging language generation as an explicit semantic supervision for facial AU recognition.
The use of local descriptions to capture intra-AU semantics and inter-AU discriminability, and global descriptions to maintain diversity across facial states is well-motivated.

Established techniques like attention mechanisms, LSTM language models, and multi-scale feature fusion are adapted appropriately.

Quantitative results demonstrate the superiority of VL-FAU over previous state-of-the-art methods in terms of AU recognition performance.

The paper is well-organized and clearly describes the motivation, approach, and experiments. The use of informative figures aids in understanding the overall architecture and workflow.

Explainable facial AU recognition has diverse applications in fields like facial expression analysis, psychology, deception detection, and human-computer interaction.

**Limitations:**

The paper primarily focuses on the AU recognition performance and provides qualitative examples of the generated descriptions. However, a linguistic analysis evaluating the quality, diversity, and coherence of the generated text descriptions is lacking.

The inclusion of multiple language models along with the vision backbone could raise efficiency and scalability concerns, especially for deployment on resource-constrained devices. The paper does not discuss computational requirements and efficiency.

The paper uses traditional LSTM-based language models for generation. However, it does not explore or compare the performance with more recent transformer-based or large language models, which could potentially improve the quality and diversity of generated descriptions.
The experiments are limited to two facial AU datasets. The evaluation could be strengthened by testing the model's generalization capabilities on facial images from more diverse scenarios.

**Suitability:**

2

---

### Official Review · Reviewer_4rXg · 2024-05-21

**Rating:** 2
**Confidence:** 3

**Summary:**

This paper studied facial action unit recognition with intrinsic relationships between different AU and explainability problems. They proposed an end-to-end Vision-Language joint learning network for explainable FAU recognition(VL-FAU), which contains a dual-level AU representation learning framework and a joint supervision method with local and global language generations.

**Strengths:**

This paper provides a novel vision-language joint learning method, which combines facial action unit recognition with language explaniability to solve the lack of explanation in existing facial action unit recognition methods. By providing explainable language descriptions at local and global levels, the VL-FAU method not only improves the accuracy of AU recognition, but also enhances the explainability of the model, which is of great significance for the application of facial expression analysis in human-computer interaction, emotion computing and other fields. Experiments are carried out on DISFA and BP4D AU datasets to verify the effectiveness of the proposed method and shows that VL-FAU has better performance than the existing methods.

**Limitations:**

There are some problems, which must be solved before it is considered for publication. If the following problems are well-addressed, the reviewer believes that the essential contribution of this paper is important for facial action unit recognition.

1. **Line 498-500**: It is not explained where the calculation of $p(y_i)$ comes from, whether it uses FAU-related features, or takes into account features that describe FAU.

2. **Line 548-549**: As shown in Equation 12, when designing the loss function, the author directly chose to add different loss functions without considering their previous coefficient relationship, which we think is not rigorous and needs to be considered.

3. **Line 564-572**: It is said here that the details of language description are not developed. We do not know whether the author's method in the training process directly uses hand-crafted data or trains according to those data. If it is the latter, the relevant details of training are not developed in paper. It is important for reviewers to evaluate the effectiveness of the method. We suggest you to put them into the main body of the paper.

4. **In Table 3**, the author compare the accuracy with the SOTA methods. This seems to be a very convincing comparison experiment, but the authors did not uniformly compare the SOTA methods(such as KS, ARR, SMA-ViT and SEV-Net) in Tables 1 and 2. In this case, we consider the experiment to be inadequate.

5. Acoording to **Line 195-197**, the authors said they extended FAU datasets with new local and global language descriptions for different facial muscle states to facilitate language-interpretable FAU recognition. However, we only saw three related examples in Section 4.4, which seemed to us more like a case study demonstrating the effectiveness of the method than an effort to expand the data set. Because we do not see any more relevant extensive data description, and the author does not have an objective evaluation standard for these AU descriptions.

**Suitability:**

2

---

### Meta-Review · Area_Chair_cDAm · 2024-07-01

**Recommendation:** Accept (Poster)
**Confidence:** 5

**Metareview:**

Reviewers 4rXg concerned about the data processing and training details, while wNWr and RifT questioned the structure and computational efficiency of the AU caption generator. Apart from the discussion of technical details, all reviewers support the novelty idea. And the rebuttal addressed most concerns


Therefore, I recommend acceptance. Please include a comparison of computational efficiency and details on constructing AU captions in the final version of the paper.

---

### Meta-Review · Senior_Area_Chairs · 2024-07-10

**Recommendation:** Accept (Poster)
**Confidence:** 4

**Metareview:**

This paper received mixed ratings initially. After rebuttal, two reviewers tend to accpt the paper while the other increased score form WA to BA. AC checked the reviews and rebuttal and recommended acceptance of the paper. SAC agrees with AC.